# Water training initiates spatially regulated microstructures with competitive mechanics in hydroadaptive polymers

Wenbo Chen [1], Caoxing Huang[1,2], Philip Biehl[1] & Kai Zhang [1] ✉

The strategy using water as a medium for dynamic modulation of competitive plasticity and viscoelasticity provides a unique perspective to attain adaptive materials. We reveal sustainable polymers, herein cellulose phenoxyacetate as a typical example, with unusual water-responsive dual-mechanic functionalities addressed via a chronological water training strategy. The temporal significance of such water-responsive mechanical behaviors becomes apparent considering that a mere 3-minute exposure or a prolonged 3-hour exposure to water induced different types of mechano-responsiveness. This endows the materials with multiple recoverable shape-changes during water and air training, and consequently even underlines the switchability between the pre-loaded stable water shapes (> 20 months) and the sequentially fixed air shapes. Our discovery exploits the competitive mechanics initiated by water training, enabling polymers with spatially regulated microstructures via their inherently distinct mechanical properties. Insights into the molecular changes represents a considerable fundamental innovation, can be broadly applicable to a diverse array of hydroadaptive polymers.

Nature fascinates with its mechanical adaptive capabilities, as evident in the remarkable stimulus-responsive behavior of various materials[1–3]. Among them, soft tissues such as cartilage and skeletal muscle stand out, as they not only offer mechanical support but also actively reconfigure themselves, adapting to their environment and performing specialized functionalities[4]. When mimicking the mechanical adaptability found in nature, a majority of the artificial materials rely on soft matrices characterized by low moduli (of kilopascals (KPa) and megapascals (MPa)). Examples of such materials include hydrogels, ionogels, elastomers, and semi-crystalline resins[5–7]. Mechanically adaptive materials with high and tunable modulus in the gigapascal (GPa) range are rare. A prime example are high-performance nanocomposites inspired by natural load-bearing structures, which benefit from the integration of mechanical adaptivity[8].

Here, we discovered previously unknown microstructures in polymers, actively regulated in space based on their distributed mechanical properties. This was achieved through an innovative water training strategy applied to cellulose phenoxyacetate (CPaE). Such a counterintuitive water training strategy can adjust specific chronological parameters for modulating the viscoelasticity and plasticity. To achieve water-responsive dual-mechanic functionalities, membrane strips of CPaE polymers are trained in water for various time periods, followed by reprogramming to air shapes prior to their immersion in water. To understand the emergent microstructure-property relationships, we attempt to assess the manipulability of the competitive mechanics and to elucidate the general principle of differential mechano-responsiveness in terms of the water permeation and plasticization at the molecular scale, which regulates the stress changes and corresponding deformation at a structural scale.

[1]Sustainable Materials and Chemistry, Department of Wood Technology and Wood-based Composites, University of Göttingen, Büsgenweg 4, D-37077 Göttingen, Germany. [2]Jiangsu Co-Innovation Center for Efficient Processing and Utilization of Forest Resources, College of Chemical Engineering, Nanjing Forestry University, 210037 Nanjing, China. ✉e-mail: kai.zhang@uni-goettingen.de

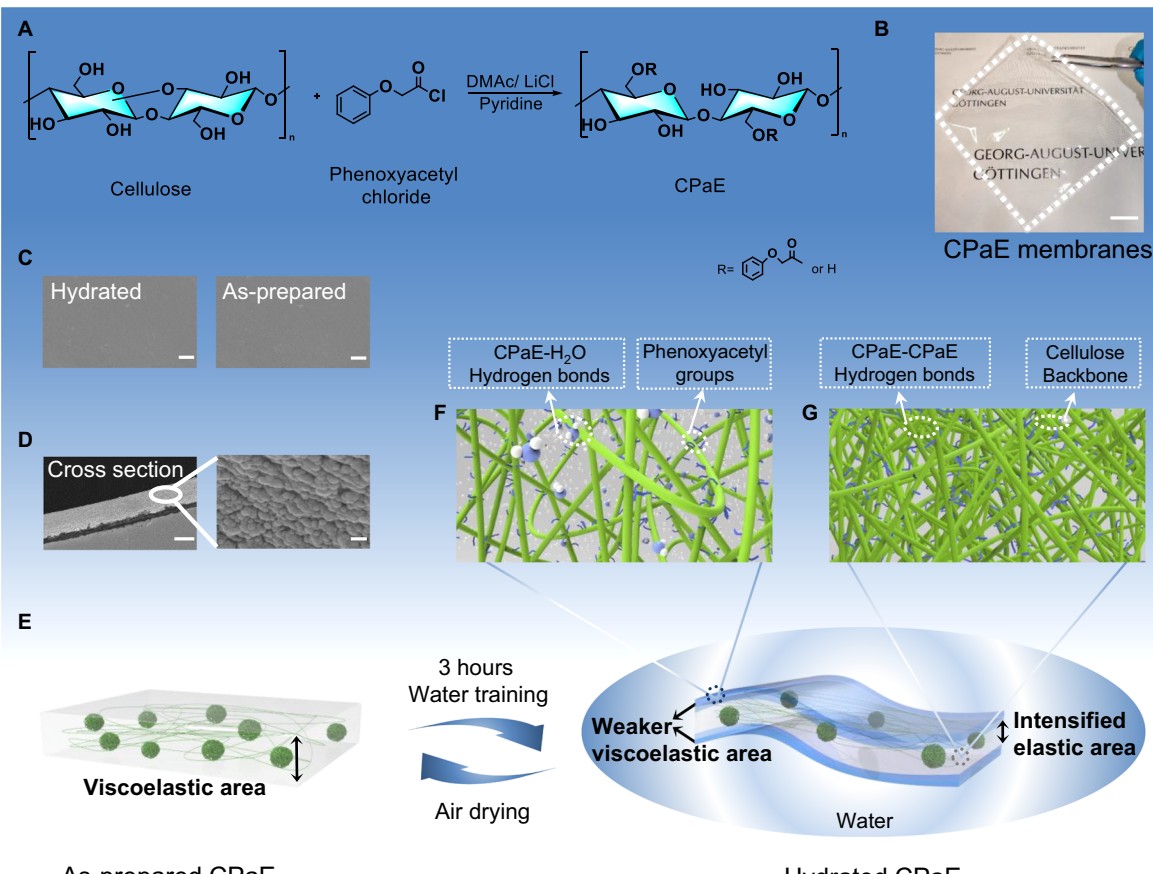

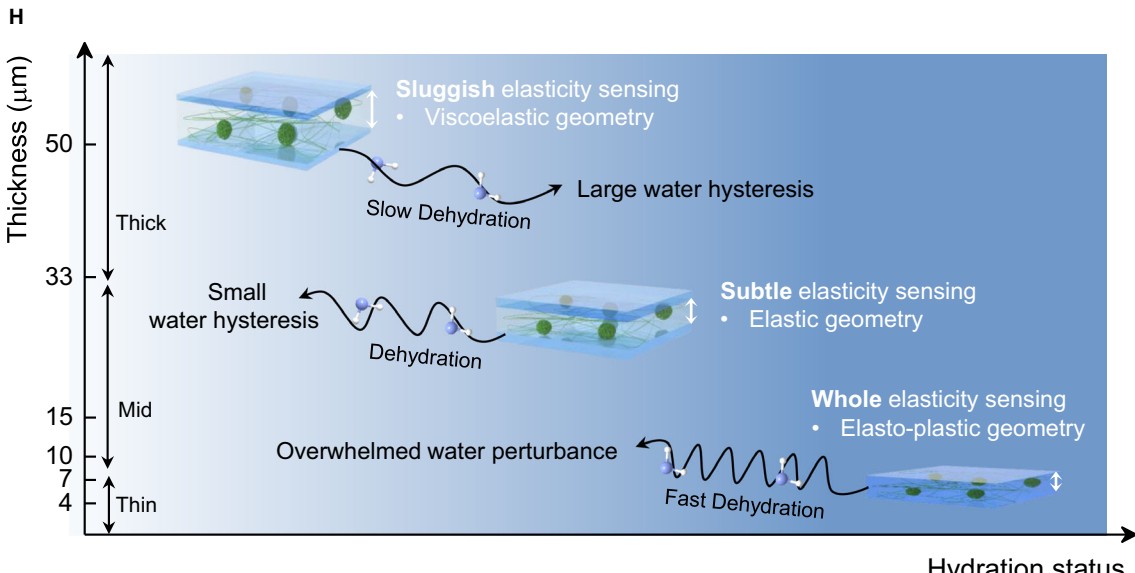

**Fig. 1 | Microstructure transformation of hydrated CPaE membranes.**
**A** Synthesis of CPaE. **B** Digital image of a transparent CPaE membrane. Scanning electron microscopy (SEM) images showing (**C**) the flat surface of hydrated and as-prepared membranes and (**D**) the cross-section of as-prepared membranes. Scale bar in (**B**–**D**) is 20 mm, 300 nm/300 nm and 10 µm/300 nm, respectively. **E** Conceptual illustration of the microstructure transformation on mid-CPaE membranes initiated by water training. Illustration shows the (**F**) intensified and (**G**) weak elastic area in hydrated mid-CPaE. **H** Schematic overview of how the microstructures of hydrated CPaE govern the competitive mechanics, including the identified area of dominated elastic behavior and the water imprint.

## Results

### Microstructure and mechanical feature

CPaE with degree of substitution of 0.25 is prepared following the synthesis approach as shown in Fig. 1A[9] (Supplementary Fig. 1, Supplementary Information). The as-prepared CPaE membranes with distinct thickness were manufactured via a facile solvent casting method. The membranes are flat and transparent with luminous transmittance of $94.3 \pm 0.3\%$ and haze of $2.0 \pm 0.1\%$ (Fig. 1B). Hydrated CPaE membranes show a comparable smooth surface to those of the as-prepared membranes (Fig. 1C). The presence of distinct

topographies of nanoparticles and continuous matrix inside the whole membrane could be explained by the phase separation during the solvent casting process (Fig. 1D).

Figure 1E illustrates the concept of water training, which initiates competitive mechanics of plasticity and viscoelasticity within CPaE membranes. Spatially separated microstructures are generated and exist as distinct areas, two outer weaker viscoelastic areas and an inner intensified elastic area. The outer areas provided plastic adaptation of CPaE polymer, whereas substantial CPaE-$H_2O$ were formed at the layered interface (Fig. 1F). The inner intensified elastic area imparts the stability for defining the initial and desired elastic geometry (Fig. 1G). The shape of the materials and their sequential alterations are determined by the relative distributions of these regions, influenced by varying hydrogen bonds between CPaE polymer chains or between the chains and water. Such competitive mechanical property can be readily adjusted by varying the distinct hydration status of membranes with different thickness (Fig. 1H). Their joint effect contributes to the dominating mechanical behavior of whole membranes and can be represented as water-responsive dual-mechanic functionalities, which in turn depend on the spatial microstructure resulting from water training.

## Demonstration of water-responsive dual-mechanic functionalities

On the basis of the dynamic competitive mechanics of plasticity and viscoelasticity within CPaE membranes, a well-defined chronological water training strategy was realized to reflect the water-responsive dual-mechanic functionalities (Figs. 1H, 2A). By fixing the membrane strips on molds and water training it for a short time period (e.g. 3 min), no water shape could be loaded (Fig. 2B). Innovatively, when extending the time period of water training (e.g. 3 h), long term stable water shapes were successfully loaded (Fig. 2C). Such water shapes exhibited high stability (>20 months) in deionized water, even in phosphate-buffered saline (PBS) solutions (Supplementary Fig. 4A, B, Supplementary Movie 1). Sequentially, an air training strategy can be further performed by fixing these wet membrane strips on molds and air-drying them in ambient conditions for 30−60 min until constant weights. Dried membranes can thus be programmed into versatile other stable air shapes, such as air helix, air ring, air alphabet and air chair, different than the water shape (Fig. 2D and Supplementary Fig. 3). Surprisingly, these air shapes, if immerged in water, can be erased and turned flat corresponding to bulk mechano-responsiveness (Fig. 2E and inset, Supplementary movie 2). Or they unusually recovered their water shapes corresponding to differential mechano-responsiveness and therefore showed the responsive shape-memory effect (Fig. 2F and inset, Supplementary movie 3−5). Such water-responsive dual-mechanic functionalities could also be introduced into other polymer materials, such as cellulose benzoate with low degree of substitution of e.g. 0.2−0.4 (Supplementary Fig. 4), but not into polymers with too strong hydrophobic properties or without differentiated stress distribution (Supplementary Fig. 5, 6). This is due to the fact that too strong hydrophobicity can lead to two troubles: 1) Water is losing its plasticizing capacity necessary to change the geometry shape of CPaE membranes; 2) The water diffusion from the surface of CPaE membrane into interior is severely restricted, thereby impeding the modulation of its inner layer's viscoelasticity. In addition, spatially differentiated stress distribution is imperative for water-responsive dual-mechanic functionalities of CPaE membranes, which is characterized by the selective formation of spatial chain relaxation. Through 3 h water training, a fraction of stress is retained via the inner layer's elasticity, where elastic memory locks the stress field. In the meantime, another fraction of stress rapidly relaxed to near 0 MPa via outer layer's hydroplasticity but establishes to a new equilibrium state after air drying, thereby advancing the heterogeneous spatial stress distribution. By contrast, homogeneous spatial stress

distribution results in a single water-responsive mechanic functionality, which is based on the bulk mechanics.

## Water training initiated dynamic mechanical behaviors and microstructure evolution

To further understand the differential mechanical properties of the spatially distributed areas, we analyzed the static mechanical properties of as-prepared CPaE membranes. Additionally, we examined these properties after distinct hydration times (Fig. 3A). By using mid-CPaE membranes with the thickness of 10−30 μm as characteristic samples, hydration time strongly and gradually affected their dynamic mechanical properties. A hydration time of 0.5 h strongly reduced the Young's modulus of hydrated mid-CPaE membranes to $1.2 \pm 0.1$ GPa and their elongation at break to $14.2 \pm 1.9\%$. Surprisingly, prolonged hydration time further to 72 h even stabilized their Young's modulus at elevated $1.7 \pm 0.1$ GPa, while the elongation at break decreased further to $9.1 \pm 1.1\%$ (Fig. 3B). In parallel, the tensile strength rapidly lowered to $68.4 \pm 1.0$ MPa after 0.5 h and maintained around 70.0 MPa for 72 h. Different than these, the fracture work of membranes after the hydration time of 0.1 h decreased constantly with longer hydration time. These results suggest that the enhanced interaction of CPaE polymer chains with water molecules substantially strengthened their total plasticity. Concurrently, the intensified elasticity of the inner layer seemingly acted to defer plastic deformation to higher stress levels. This finding is in agreement with the fact that water plasticization leads to the substantial elimination of entanglements in amorphous polymers[10] and thus endows the CPaE membranes with hydroplastic properties. In parallel, the persistence of trapped entanglements represented as the inner elastic layer can still contribute to the mechanical resilience, as elucidated by Rubinstein and Colby[11]. For instance, both the hydrated thick-CPaE membranes and hydrated thin-CPaE membranes exhibited identical initial linear elasticity to their respective as-prepared membranes (Supplementary Fig. 7). Remarkably, the as-prepared and hydrated CPaE membranes of various thickness showed comparable or superior tensile strengths and Young's modulus even after several regeneration cycles (Supplementary Fig. 8), compared with currently widely used thermoplastic plastics, such as PE and PP, as well as thermosetting materials including epoxies and phenolics[12] (Fig. 3C).

The stress relaxation of hydrated CPaE membranes at in situ cyclic RH between 95 and 30%, provided insights into the complex strain and stress distribution, as well as stress transmission in response to the microstructure change (Fig. 3D). While the stress in hydrated mid-CPaE membranes at 95% RH decreased from maximally 18.0 MPa to 4.9 MPa, the elastic inner layer was major involved and retained a considerable portion of stresses. By lowering the RH to 30%, the stress recovered approximately to the original level results in the majority of stress accumulation and turns the outer layer to weaker viscoelastic state. This indicated the presence of concealed spatially distributed micro-structures characterized by an uneven distribution of strain while changing the RH[13]. Upon exposure to 95% RH again, water molecules abruptly penetrate into the membranes from the outer layer, and thus to relax the accumulated stresses to near 0 MPa rapidly. This further indicates that the adsorption and desorption of water molecules occurred in the same outer layer, which is a typical feature of reversible hydroplasticity. The statement of a distinctive spatial stress distribution in hydrated mid-CPaE membranes is strongly evidenced by the facts: (1) The prominent difference in the stress relaxation behavior between the cycle 1 and subsequent cycles; (2) The Polarized Optical Microscopy (POM) images on the cross section of air dried CPaE membranes being made after 3 min and 3 h water training, respectively (Supplementary Fig. 9). In addition, variable surroundings with changing RH altered the applied strain contributed to the non-linear elasticity of hydrated CPaE, while limited strain change still restricted the range of possible movements or deformations within the material. The

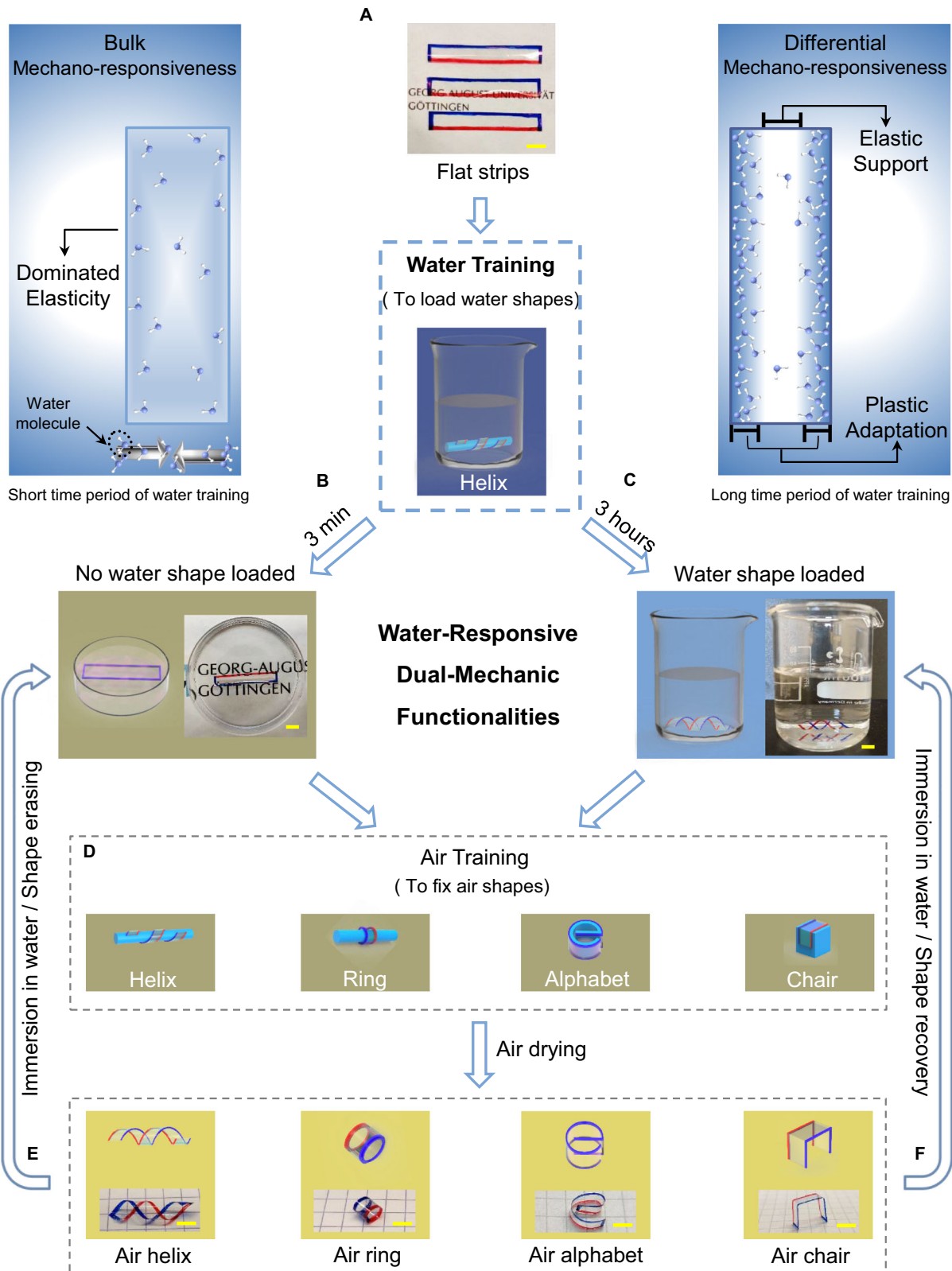

**Fig. 2 | Water-responsive dual-mechanic functionalities of CPaE membranes.**
**A** Schematic of loading mid-CPaE membrane strips with water shapes via water training and their water-responsive dual-mechanic functionalities. Flow diagram showing the water training process for (**B**) short time period of 3 min and (**C**) long time period of 3 h. Water helix was chosen as a representative shape for clarity. See more cases in the Supplementary Fig. 2. **D** Schematic of processing wet membrane strips into distinct air shapes via air training strategy. **E**–**F** Flow diagram showing the bulk and differential mechano-responsiveness of the generated air shapes to water. The corresponding behaviors are (**E**) shape erasing and (**F**) shape recovery, respectively. Insets near (**B**) and (**C**) show the mechanical features as either bulk or differential mechano-responsiveness with corresponding water distribution over the cross-section of the membranes after a short or long time period of water training. The edges of membrane strips ($30 \times 5 \, mm^2$) were marked for clarity. Scale bars, 5 mm.

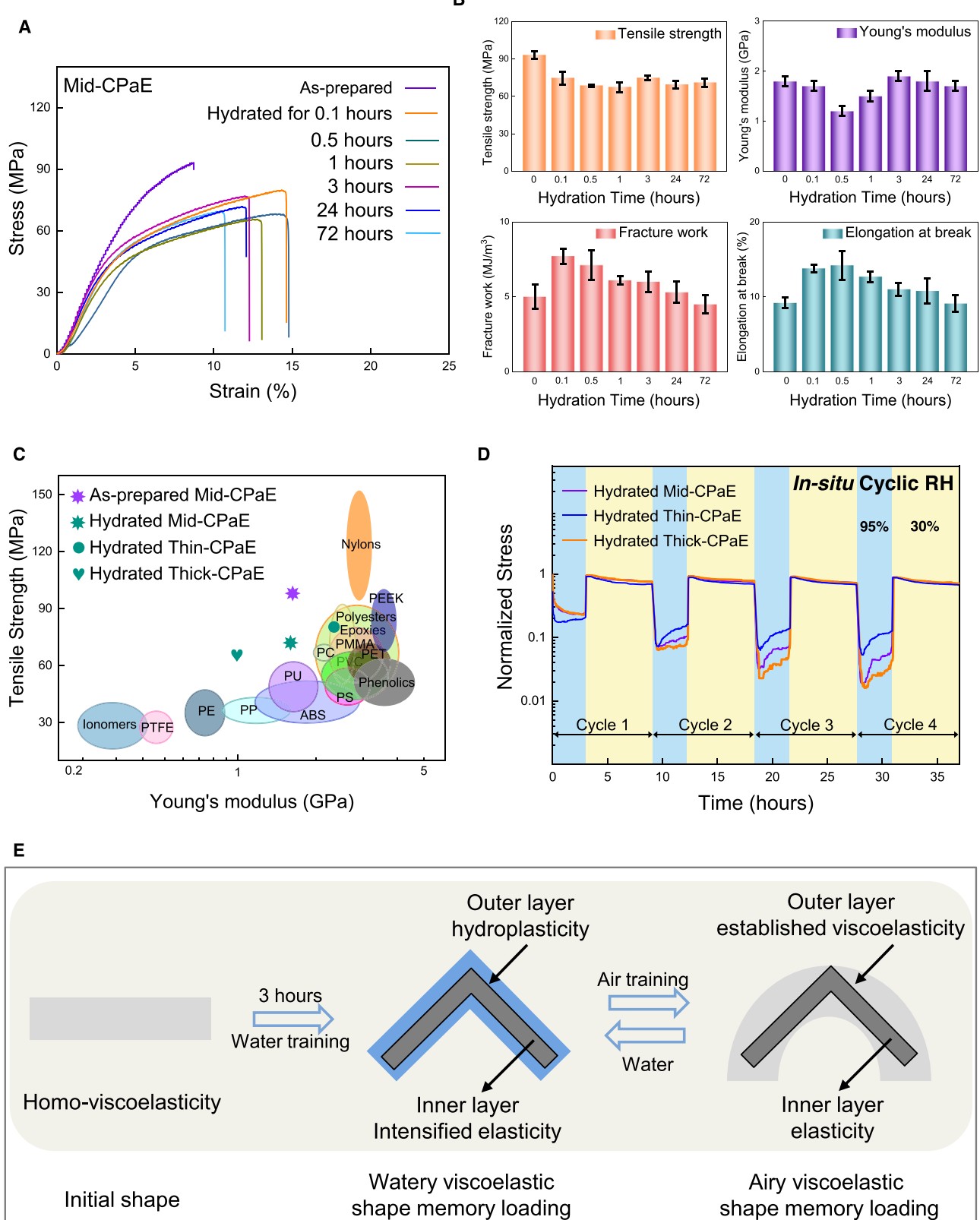

**Fig. 3 | Mechanical behaviors of CPaE membranes interacting with water.**
**A** Stress–strain curves of as-prepared mid-CPaE membranes and after hydrated for distinct time periods. **B** Calculated tensile strength, Young's modulus, elongation at break and fracture work. 0 min means the as-prepared membranes. Error bars are SD. **C** Ashby diagram of tensile strength vs. Young's modulus for as-prepared mid-CPaE and hydrated (hydrated for 72 h) CPaE membranes compared with typical thermoplastics and thermosetting plastics[12]. **D** Normalized stress relaxation of hydrated thin, mid and thick-CPaE membranes under an in situ cyclic relative humidity (RH) between 95 and 30%. Thin, mid and thick-CPaE indicates the membranes thickness range from thin, middle to thick. **E** Visual demonstration of watery and airy viscoelastic shape memory in CPaE membranes, distinguished by its spatially distributed mechanical feature. Homo-viscoelasticity means homogenous viscoelasticity.

specific driving factors behind this process, whether it is entropic or enthalpic, remain unclear[14,15]. Interestingly, the nonlinear elasticity exhibited not only a response to stress relaxation over time, but also an instant correlation with water-induced hydroplasticity at 95% RH and variable mechanical cycling between 95 and 30% RH. To conclude, the nonlinear-elasticity of hydrated mid-CPaE was predominantly attributed to the existing elastic inner layer, and absorbed the majority of recoverable strain, whereas the external layer afforded plasticity and acted as stress transfer. The presence of such nonlinear-elasticity, although with different design, also exists in natural active biopolymer networks to regulate the mechanical properties of cells[16]. Herein, the water-induced plastic perturbation caused challenges in bridging the gap between macroscale mechanical behaviors and microstructure change, which exist for other water-responsive materials[16–18]. In this respect, we focused on assessing stress relaxation in membranes of different thicknesses, considering the substantial role of water permeation and diffusion on the plasticity.

Hydrated thin-CPaE membranes demonstrated the fastest stress relaxation with minimal residual stress compared to hydrated mid and thick-CPaE membranes. This suggests that thinner membranes of 4−6 μm could accelerate the overall progression of plasticity due to the water absorption at high RH, facilitating the quick conversion of elastic zones to plastic zones within the materials with simultaneous transfer of local strain to global strain. As the result, the difference between inner and outer layer was very small. The corresponding creep behavior was consistent with this result (Supplementary Fig. 9A). On the contrary, hydrated thick-CPaE membranes of 48−50 μm did not reach equilibrium within 3 h, implying that complex inner stress gradient was formed in the materials. As for all three different thickness, each material showed an increase of stress at high RH of 95%, which could be explained by the strain hardening effect based on the prolonged plasticity excitation. Decreasing RH to 30%, stress reconstruction induced by water desorption primarily occurred at the outer layer. No substantial difference in stress relaxation was observed in each material, attributed to the strong interactions among CPaE polymer chains at 30% RH. Compared to lower thickness, the residual stress of thick-CPaE remained nearly constant. This suggested that higher thickness reduced the development of plasticity due to their lower overall contents, while the inherent softer elasticity within the inner layer made the internal chain segments prone to sliding during the RH cycle, resulting in an irreversible deformation. Decreasing the overall contents of plasticizable regions, competitive plastic and elastic regions will endow the materials with dual mechano-responsiveness. Dynamic changes in the mechanical properties of CPaE membranes in watery and airy states, resulting from competing mechanical forces were demonstrated in Fig. 3E. During 3 h water training, the intensified elasticity characterizing the viscoelasticity of inner layer and the hydroplasticity of outer layer engaged in a fierce competition, with the former maintaining its elastic geometry and the latter restoring its original geometry. The inner layer serves as a baseline stress level while the outer layer showed thoroughly stress relaxation due to sufficient bounded water molecules (Supplementary Fig. 10). A delicate balance of plasticity and viscoelasticity manifested itself in an elastic shape memory response, presenting a watery viscoelastic shape memory. Upon air drying, new viscoelastic geometry was established in the outer layer due to the reversible hydroplasticity. Airy viscoelastic shape memory maintained when the outer layer's newly established viscoelasticity surpasses that of the inner layer. This shape memory promptly restored to its original geometry upon water sorption. Notably, the viscoelasticity of the inner layer began to impede the development of hydroplasticity as water diffuses until it reaches its boundaries, thereby preserving the elastic geometry. To further elucidate the stress transfer and deformation change from such competitive mechanics, we visualized such nonlinear elastic behaviors on the whole timescale by using the modified viscoelastic-plastic model

consisting of an elastic-plastic branch and a viscoelastic branch[19] (Supplementary Fig. 11).

With the presence of spatially distributed regions with distinct mechanical properties, the correlation between the microstructure formation of hydrated CPaE and the state of water molecules was verified by DMTA (Fig. 4A). The storage modulus of the hydrated mid-CPaE membranes decreased gradually from $6.01 \pm 0.49$ to $4.69 \pm 0.54$ GPa in the temperature range of −80 to −15 °C and intensely to $3.05 \pm 0.40$ GPa at 35 °C, while the loss modulus decreased gently from $0.43 \pm 0.02$ to $0.30 \pm 0.02$ in the temperature range of −80 to 35 °C. Correspondingly, the damping factor decreased slightly from $0.072 \pm 0.002$ to $0.063 \pm 0.006$ with increasing temperature from −80 to −15 °C, and then abruptly rose to $0.097 \pm 0.003$ at 35 °C. Moreover, an apparent broad peak of damping factor in the range of −15 and 60 °C was observed. These results suggest that water molecules adsorbed inside the hydrated mid-CPaE should be in a highly restricted state, because of the following aspects: 1) the formation of massive strongly bonded $CPaE-H_2O$ reduced the mobility of water molecules; 2) the enhanced hydrogen-bond sites of $CPaE-H_2O$ hinder the nucleation of ice crystals. As the temperature rose to −15 °C, the storage modulus rapidly decreased due to the increased molecular mobility on the confined surface of mid-CPaE, followed by a slight increase due to the water evaporation with rising temperature to over 35 °C. This trend indicates the competitive mechanics during dynamic water variation process that further results in a substantial microstructure transformation. The latter can be distinguished by 1) the robust mechanical performance in low temperature range; 2) a trade-off mechano-responsiveness in the whole temperature range.

RH hysteresis of as-prepared and hydrated CPaE provided further information about the different hydrogen-bond networks. As shown in Fig. 4B, the storage modulus of as-prepared mid-CPaE decreased from $4.86 \pm 0.08$ GPa to $2.44 \pm 0.06$ GPa with increasing RH from 18 to 92%. With in situ decreasing RH from 92 to 18%, the storage modulus became first slightly lower and then restored to $4.48 \pm 0.09$ GPa at 18% RH. Correspondingly, the damping factor rose from $0.071 \pm 0.002$ to $0.134 \pm 0.003$ in the RH range of 92 to 18% and then decreased to $0.081 \pm 0.002$ at 18% RH. As comparison, the storage modulus of hydrated mid-CPaE gradually increased from $2.09 \pm 0.07$ GPa to only $3.90 \pm 0.17$ GPa with decreasing RH from 92 to 18% (Fig. 4C). Particularly at the RH from 80%, it began to drastically decrease and reached $2.47 \pm 0.09$ GPa at 92% RH. These results imply that water molecules were abruptly concentrated at the $CPaE-H_2O$ binding sites at high RH of more than 80%. Obviously, hydrated CPaE showed substantially different hydrogen bonds as distributed between CPaE-CPaE and $CPaE-H_2O$ than those in the as-prepared CPaE. Several reasons could account for this result. On the one hand, only a small portion of CPaE-CPaE hydrogen bonds in hydrated CPaE could be reformed and replaced by $CPaE-H_2O$ hydrogen bonds, which requires further energy for water binding and penetration. On the other hand, the variations in sizes and relative positions of hydrophobic and hydrophilic regions along the polymer backbone[20] make the interactions of the surrounding water molecules with hydrated CPaE more complex. Moreover, by evaluating the storage/loss modulus and damping factors as a function of temperature, frequency and RH, the dynamic viscoelasticity behaviors of mid-CPaE membranes was modularly affected by the water-induced plasticity (Supplementary Fig. 12). In comparison, their storage and loss modulus remained stable at constant RH even after 4 h at 10 Hz or between 0.1 and 100 Hz (Supplementary Fig. 13A, C). Additionally, after 3−9 cycles of water and air training strategy, the hydrated mid-CPaE showed similar storage and loss modulus of $2.27 \pm 0.25$ and $0.25 \pm 0.02$ GPa after 4 h at 95% RH, respectively (Supplementary Fig. 13B, D). Therefore, more cycles should not induce further substantial changes in the mechanical properties of CPaE membranes. Therefore, the strong interactions among CPaE polymer chains, both in air and water, significantly contribute to water binding,

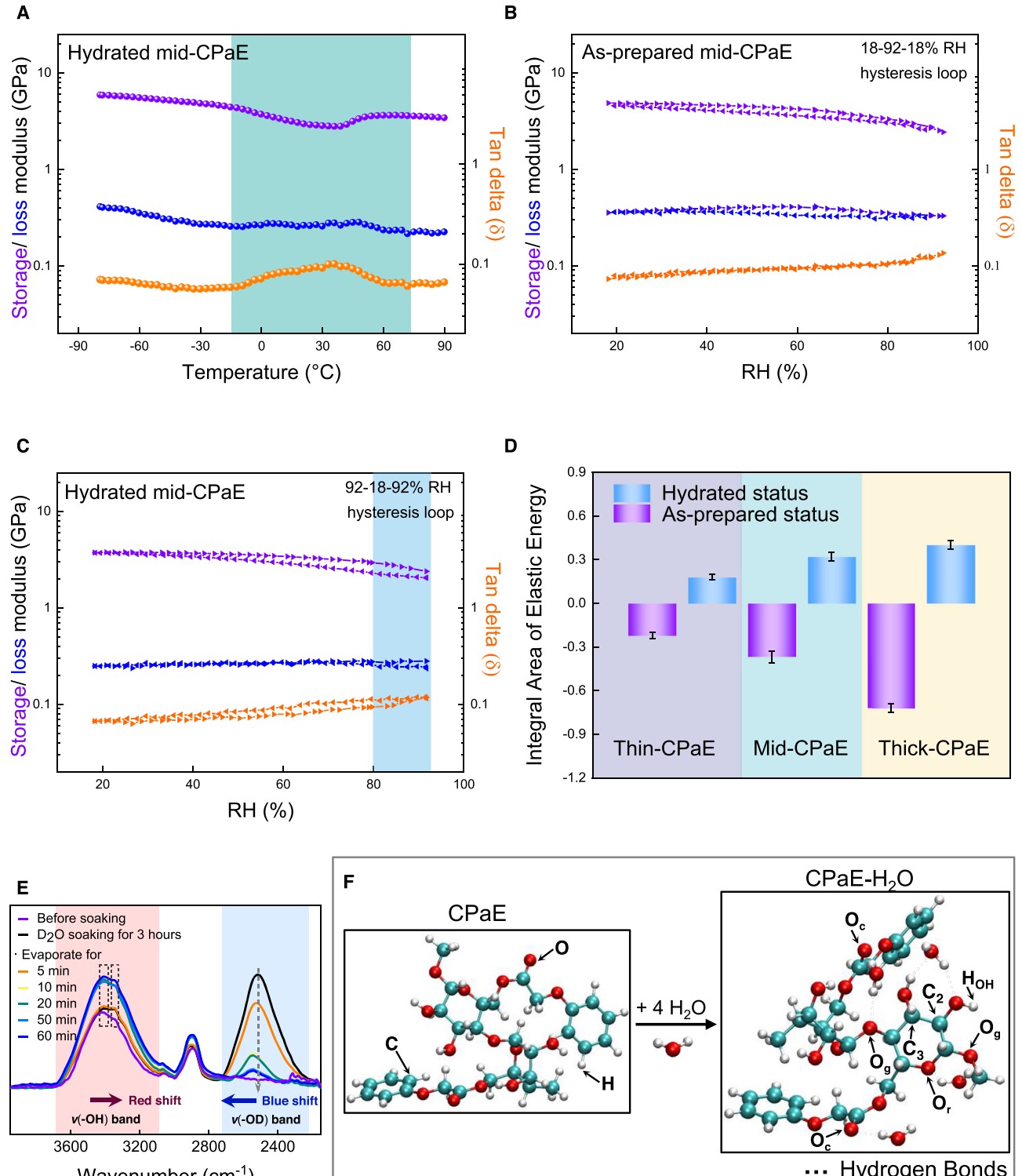

**Fig. 4 | Water determines the microstructure evolution in CPaE membranes.**
**A** Dynamic mechanical thermal analysis (DMTA) of hydrated mid-CPaE from −80 to
92 °C at 10 Hz. Highlighted region shows the abrupt change. RH hysteresis loop
begins at (**B**) 18% RH for as-prepared mid-CPaE and (**C**) at 92% RH for the hydrated
mid-CPaE. RH hysteresis loop data of thin and thick-CPaE are to be found in Sup-
plementary Fig. 12C–F. **D** Integral area of elastic energy of as-prepared and
hydrated thin, mid and thick-CPaE based on the RH hysteresis loop. Thin, mid and
thick-CPaE indicates the membranes thickness range from thin, middle to thick.

Error bars are SD. **E** ATR-FTIR spectra of the CPaE membranes soaked in $D_2O$ for 3 h,
and after various time intervals of exposure to the ambient condition (26.5–27.4 °C,
32–37% RH). $v$(OH)/$v$(OD) stands for the -OH/-OD characteristic band. **F** Optimized
ball-and-stick model plots of CPaE and CPaE-$H_2O$ based on atoms in molecules
theory. Binding energy of the complexes computed at the CCSD(T)/jul-cc-pVTZ
level. $O_c$, $O_r$ and $O_g$ denoted the carbonyl oxygen from phenoxyacetyl group,
oxygen on glucose ring and glycosidic linker, respectively.

resulting in the spatial distribution of stress and thereby enabling water-responsive dual-mechanic functionalities.

In addition, the elastic energy variations extracted from the RH hysteresis loop provided information about the water imprints in as-prepared and hydrated CPaE (Fig. 4D). The energy dissipation of the as-prepared CPaE indicates the water-retaining capacity of the materials upon desorption. For example, the thick-CPaE showed the largest energy dissipation area of $0.72 \pm 0.03$, resulting in a large water hysteresis and slow dehydration process. This suggested that more water molecules remained attached to the polymer chains compared to the thinner samples, so that much lower RH was required for the same storage modulus. By contrast, lower elasticity energy of the hydrated CPaE indicates the weaker binding capacity between CPaE polymer chains and water upon sorption. For example, the thin-CPaE showed the smallest elasticity energy area of $0.18 \pm 0.02$, resulting in a minimal binding hindrance to water molecules. Therefore, the presence of the larger water hysteresis or perturbation demonstrates more difficult disruption of the further elasticity in the inner layer and therefore its gradual intensified ability to resist plastic deformation due to water binding, which is detrimental for water-responsive dual-mechanic functionalities.

ATR-FTIR spectra by the dynamic $D_2O/H_2O$ exchange were recorded to trace the molecular change between CPaE polymer chains and water molecules during the microstructure transformation in the CPaE membranes. Figure 4E demonstrates the obvious change in the -OH and -OD peaks after soaking the membranes in $D_2O$ for 3 h by showing a strong peak for $\nu(OD)$. During evaporation for 60 min, the intensity of the $\nu(OD)$ band decreased due to the evaporation of free $D_2O$, while the intensity of the $\nu(OH)$ band increased because of the formation of more hydrogen bond between O of remaining $D_2O$ and H of CPaE. This suggests that the strong potential of forming more hydrogen bonds with introduced water, which will lead to modifications in molecular cooperativity and change the original hydrogen-bond pattern[21]. Quantitative hydrogen bonding energy of CPaE-$H_2O$ was fitted by calculating the critical point electron density via atoms in molecules theory (Supplementary Table. 1). The typical hydrogen bonds between CPaE and $H_2O$ involved one single and three double hydrogen bonds[22]. The formation of double hydrogen bond complexes facilitated the creation of an intensified network, as they acted as bridges between CPaE polymer chains. Importantly, the occurrence of this phenomenon was found to depend on the duration of exposure of the materials to water, as shown above during the $D_2O$ exchange[23]. Moreover, the Fig. 4F illustrates the optimized structure of the minimum point on the potential energy surface of CPaE and CPaE-$H_2O$, which is expected to be consistent with the conformational changes that take place within the CPaE matrix.

## Discussion

In summary, we revealed water-responsive dual-mechanic functionalities in a group of polymers derived from sustainable materials, specifically cellulose. This was achieved through the development of an innovative, counterintuitive, and eco-friendly water training approach that induced dynamic changes in the plasticity and viscoelasticity of the materials. On the one hand, bulk mechano-responsiveness was characterized by the memory of their initial microstructure. On the other hand, differential mechano-responsiveness was characterized by a microstructure transformation in an aqueous environment and the presence of spatially mechanically different regions that allowed water-induced multiple shape changes.

The notion is further enhanced that CPaE polymers are a distinct class of novel plastics beyond hydroplastics, thermoplastics and thermosets. Notably, our findings have gone way beyond the passive goal of solving the plastic pollutions, and turned to introduce a promising perspective for the creation of emergent plastics with actively spatially regulated microstructures. The resulting water-responsive dual-mechanic functionalities pave a promising avenue for both scientific research and manufacture practices by using emergent plastics derived from sustainable compounds. Moreover, these renewable, recyclable and actively spatially regulated microstructures exhibit significant mechanical properties both in water and air. It is foreseeable that the considerable advancement can offer promising prospects for extending their lifespan across various areas, including but not limited to medical devices, flexible electronics and soft robotics, ultimately realizing a fully sustainable circularity.

## Methods

### Synthesis of CPaE

CPaE with low DS were synthesized via homogeneous acylation of cellulose as follows. Briefly, 1 g of dried microcrystalline cellulose (particle size of 50 μm, SERVA-Electrophoresis) and 40 ml of DMAc (99.5%, Sigma–Aldrich) were added to a 100 ml three-necked flask with a magnetic stir bar and attached a condenser. The mixture was stirred at 130 °C for 30 min, followed by the addition of 2.8 g of LiCl (MP Biomedicals) at 100 °C. Under continuous stirring, the solution was allowed to cool down to room temperature overnight, leading to a clear solution. Thereafter, the solution was heated to 60 °C, before phenoxyacetyl chloride (0.77 ml) (98.0%, Sigma–Aldrich) and pyridine (0.87 ml) (99.0%, Sigma–Aldrich) were added to produce CPaE. The reaction was carried out by maintaining the temperature at 60 °C while stirring for 3 h. The mixture was subsequently precipitated in 200 ml of methanol (99.8%, TH. Geyer). The product was collected by centrifugation, purified by repeated precipitation in methanol and dissolution in DMSO (99.5%, Sigma–Aldrich).

### Preparation of CPaE membranes

CPaE membranes were made by using solvent casting technique. The wafer substrate ($55 \times 55 \times 0.7$ mm³) was sonicated twice at 37 Hz, 65 °C in DMSO and deionized water for 3 min, respectively, followed by blow-drying with nitrogen. Thereafter, it was sandwiched between two teflon blocks. The blocks were $90 \times 90 \times 15$ mm³ in size and the upper part owns a hollow cylinder ($\varnothing = 50$ mm, $h = 15$ mm) at the center, as well as threads on the margin for adjusting the level and assembly fixation. Following 5 ml of CPaE solution was transferred into the assembled molds at a concentration of 10 mg ml$^{-1}$. After drying at 80 °C and 15 mbar in a vacuum oven for overnight, CPaE membranes were obtained.

### Scanning electron microscopy (SEM)

The images of cross-section and surfaces of CPaE membranes were measured using an SEM Leo SUPRA 35 Instrument (Carl Zeiss SMT GmbH). A 10-nm layer of carbon was vacuum-coated on the samples before observation.

### Nuclear magnetic resonance (NMR) spectroscopy

$^1$H (frequency of 500 MHz) and $^{13}$C (frequency of 125 MHz) NMR spectroscopy of CPaE in deuterated DMSO were recorded with a Bruker DRX 500 spectrometer (Bruker, BioSpin GmbH). The repetition delay was 5 s. A total of 65 and 16,000 scans were collected for $^1$H and $^{13}$C NMR spectroscopy, respectively.

### Elemental analysis

Elemental analysis was performed on an elemental analyser Vario EL III CHN instrument from Elementar (Hanau).

### Fourier-transform infrared (FTIR) spectroscopy

FTIR spectroscopy was conducted on BRUKER ALPHA Spectrometer (Bruker, Germany) at room temperature between 4000 and 400 cm$^{-1}$ with resolution of 4 cm$^{-1}$ using Platinum ATR. CPaE membranes were measured twice per 24 scans and average spectra were generated. The samples were dried under vacuum at 105 °C for 60 h and blow dried a further 3 min under nitrogen gas before test. The results collected

before and after soaking in $D_2O$ were normalized by the intensity of the peak at $892\,cm^{-1}$, which corresponds to the asymmetric stretching vibration of C-O-C bond in the amorphous region.

## Static mechanical measurement

Mechanical tests were performed on a Z3 micro tensile test machine with a 50 N load cell (Grip-Engineering Thümler GmbH). The CPaE membrane strips with a dimension of $30 \times 5\,mm^2$ were loaded into the test machine with a clamp distance of 8.0 mm and subjected to uni-axial extension with a constant rate of $2\,mm\,min^{-1}$ until rupture. In each case, three to five equal samples were measured to ensure the accuracy of the results. The stress was calculated by dividing force by cross-section area, while the strain (%) was defined as $(L - L_O)/L_O \times 100\%$, where $L$ is the instantaneous length and $L_O$ is the initial length of the specimen. Toe compensation was performed before any calculation to obtain correct values by using stress–strain curves. The calculation of tensile strength, Young s modulus, elongation at break and fracture work, as well as the toe compensation were performed according to ASTM D882-02[24].

## Dynamic mechanical thermal analysis (DMTA)

DMTA measurements were carried out on DMA GABO EPLEXOR system (NETZSCH GABO Instruments GmbH) with a force sensor of 50 N. The membrane strips with dimensions of 30 mm × 5 mm were loaded into the machine with a clamp distance of $8.0 \pm 0.2$ mm. The measurements were executed with a contact force of 0.5 N, a static strain of 0.5%, and a dynamic strain of 0.1%. Stress relaxation behaviors of hydrated CPaE membrane strips (hydrated for 24 h) were measured at 25 °C with a constant strain of 2% under constant RH of 95% and subsequently under in situ cycling RH with repeating RH change between 30% (cycling time of 3 h) and 95% (cycling time of 6 h). Before testing, the membrane strips were preconditioned at 25 °C and 95% RH for 15 min. Temperature sweep tests were measured with temperature increasing rate of $3\,°C\,min^{-1}$ and frequency of 10 Hz. Cyclic humidity sweep tests were performed with RH increasing/decreasing rate of 2% $min^{-1}$ at 25 °C and frequency of 10 Hz. Creep behaviors of hydrated CPaE membrane strips (hydrated for 24 h) were carried out at 25 °C with a constant stress of 5 MPa under constant RH of 95% for 2 h). Before testing, the membrane strips were preconditioned at 25 °C and 95% RH for 60 s. All measurements were carried out within the linear viscoelastic region of CPaE membranes (Supplementary Fig. 13). For each case, three to five parallel tests were performed to ensure the accuracy of the collected data.

## All-electron DFT calculations based on atoms in molecules (AIM) theory analysis

All-electron DFT calculations have been carried out by the latest version of ORCA quantum chemistry software (Version 5.0.3)[25]. For geometry optimization calculations, the corrected version of r2SCAN exchange-correlation functional proposed by Grimme (so-called r2SCAN-3c) was adopted[26]. The singlet point energy calculations were performed with B3LYP functional and ma-def2-TZVPP basis set[27]. The DFT-D3 dispersion correction with BJ-damping was applied to correct the weak interaction to improve the calculation accuracy[28]. Electron density analysis of Bond critical point (BCP) was performed by Multiwfn software[29].

## Luminous transmittance, Haze

The luminous transmittance and haze of CPaE membranes was measured on the transparency Test System & Haze Meter Model of haze-gard i from BYK (Emmeram Karg Industrietechnik). The membrane surface is illuminated perpendicularly and the transmitted light is measured photo-electronically using an integrating sphere (0°/diffuse geometry). CPaE membranes were prepared in a dimension of $10 \times 10\,mm^2$. All the measurements were tested for five times

at ambient environment. The results of luminous transmittance and haze, as well as calibration were performed according to the ASTM-D1003.

## Contact angle (CA)

The surface wetting properties of the specimens were assessed on a Drop Shape Analysis System (DSA 25E, Krüss) at 20 °C, 60% RH. The static contact angle value of Milli-Q water and diiodomethane were measured by using the sessile droplet method with a dosing volume of 1.0 μL and a dosing rate of 1.0 μL/s. All measurements were performed at least three times, and the static contact angle was acquired by taking the mean from more than 10 equivalent measurements. CA were calculated with the Young-Laplace equation.

## Data availability

The data that support the plots within this paper and other findings of this study are available from the corresponding author upon request. Source data are provided with this paper.

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

## Acknowledgements

K. Z. thanks the German Research Foundation (DFG) and Lower Saxony Ministry of Science and Culture for the project INST186/1281-1/FUGG. W. C. thanks the China Scholarship Council for the PhD grant. C. H. thanks Alexander von Humboldt Foundation for financially supporting his Postdoc Fellowship (Ref 3.5-1221348-CHN-HFST-P). We thank Dr. Lukas E. from University of Göttingen for supporting the DVS measurements. We thank Ying Z. from University of Göttingen for her help on editing movies.

## Author contributions

K. Z. developed the concept and W. C. designed the experiments. K. Z. supervised the project. W. C. conducted the experiments with the assistance of C. H. and P. B. The data were analysed and processed by W. C. and K. Z. W. C. and K. Z. prepared the original manuscript and all authors contributed to review and editing.

## Funding

## Competing interests

The authors declare no competing interests.
