## [Peer Review File · Nature Communications]

REVIEWER COMMENTS

Reviewer #1 (Remarks to the Author):

This work developed an interesting counterintuitive and eco-friendly water-training strategy which induces dynamical modulation of plasticity and viscoelasticity in the materials. Different from previous hydroplastics which mainly used water for passive processing, this work attempted to employ water to actively regulated the shape recoveries of the formed sustainable material. The authors have given an exhaustive characterization and a sound explanation for the observed phenomenon. Therefore, I am glad to support the publication of this work in Nature Communications. However, a major revision is still needed to further polish their findings.

1. I do not recommend to use the term “mesostructure” to describe the formed spatially distributed layers in this material. “Mesostructure” is usually employed to describe the structures at the mesoscale, which obviously do not apply to the situation in this work.

2. How to understand the relationship between the phase-separated structure (Figure 1D) and the water diffusion in the hydrated CPaE? It seems water may probably diffuse in the boundaries of nanophases. If so, the formation of viscoelastic state may have an origin at the micro and meso scales.

3. The stress distribution in the cross-section of water-trained and air-dried films may be observed by polarized optical microscopy to understand the different shape recovery pathways in different cases.

4. In Figure 3B, “Fracture energy” should be “Fracture work” or “Toughness”. “Fracture energy” generally describes the resistance to crack propagation and has a typical unit of “J m⁻²”.

5. In the stress relaxation tests, the authors wrote, “Starting at 95% RH, the initial stress of hydrated mid-CPaE significantly and quickly decreased from maximally 18.0 MPa to and remained less than 0.1 MPa at 95% RH in the following cycles.” Such a substantial stress relaxation corresponds to the complete plasticization of the materials. However, even immersed in water for 72 h, the material is still very stiff and strong (Figure 3a). How to explain this difference?

6. The author claimed that the water-responsive dual-mechanic functionalities is dominated by dynamical modulation of plasticity and viscoelasticity in the plastic with water training. However, the water-modulated plasticity and viscoelasticity of cellulose phenoxyacetate is not clearly demonstrated in the manuscript.

7. Water content in the hydrated cellulose phenoxyacetate is an important parameter in determining the shape change behavior. What’s the water content of the hydrated cellulose phenoxyacetate after immersing in water for different time? In Figure 2A, it is demonstrated that water molecules distribute inhomogeneously in the strip. Is there any way to detect the water distribution in the cellulose phenoxyacetate at various hydration time?

8. It is mentioned in Page 7, Line 132-136 that “Such water-responsive dual-mechanic functionalities could also be introduced into other polymer materials..., but not into polymers with too strong hydrophobic properties or without differentiated stress distribution (fig. S5 to 6).” More explanation needs to be provided.

Reviewer #2 (Remarks to the Author):

In the work, "Water training initiates spatially regulated mesostructures with competitive mechanics in hydroadaptive polymers" W. Chen et al reported the study of polymeric membranes with complex shape memories. The membranes can restore water-trained shape after sequential air training. A substantial part of the manuscript is devoted to the study of various elastic properties of these materials. However, some issues should be resolved before I can recommend this manuscript for publication.

1. The motivation for this work is unclear to me. The authors started with the problem of global plastic pollution, unfortunately, for me, it remains unclear how these materials could help in solving the problem.
2. I suggest expanding the discussions on the origin of the revealed memory-shape properties.
3. There are several grammatical and stylistic errors throughout the main text. A thorough revision of language and grammar is necessary.
4. I haven't found the input files for the quantum chemistry calculations. Please provide them as supplementary materials.
5. In Fig. 3E the authors demonstrate the model diagram, however, I didn't find the calculations in the text. How one can verify the applicability of the model?

Responses to Reviewers' Comments and Revisions Made

(Blue: Reviewers' remarks; Black: Our response)

(The changes are highlighted in yellow in the main text and supporting information.)

Reviewer #1

This work developed an interesting counterintuitive and eco-friendly water-training strategy which induces dynamical modulation of plasticity and viscoelasticity in the materials. Different from previous hydroplastics which mainly used water for passive processing, this work attempted to employ water to actively regulated the shape recoveries of the formed sustainable material. The authors have given an exhaustive characterization and a sound explanation for the observed phenomenon. Therefore, I am glad to support the publication of this work in Nature Communications. However, a major revision is stilled needed to further polish their findings.

Response

We appreciate Reviewer #1 for recognizing the importance and novelty of our research work. We have considered these insightful concerns carefully and have endeavored to address these constructive comments properly. Please find our point-to-point response below.

1. I do not recommend to use the term "mesostructure" to describe the formed spatially distributed layers in this material. "Mesostructure" is usually employed to describe the structures at the mesoscale, which obviously do not apply to the situation in this work.

Response

We appreciate reviewer#1's suggestion regarding the terminology used to describe the spatially distributed layers in our material. We agree that "mesostructure" cannot accurately convey the nature of the layers formed in our work. In response to this point, we have replaced "mesostructure" with "microstructure" in the revised manuscript.

2. How to understand the relationship between the phase-separated structure (Figure 1D) and the water diffusion in the hydrated CPaE? It seems water may probably diffuse in the boundaries of nanophases. If so, the formation of viscoelastic state may have an origin at the micro and meso scales.

Response

We appreciate reviewer#1 for raising the insightful point. We are delighted that Reviewer#1 fully recognized the novelty of our work. Worthy to mention that it is decent to use spatially distributed microstructures with differential mechanical attributes rather than phase-separated structures for describing the structure characteristic in CPaE membranes. This is because phase separation is unlikely to occur in our material (with 15% of maximum water content, according to DVS result), which is in contrast to that often observed in water-rich systems, such as hydrogels.

Herein, we propose that water diffusion induces a spatially distributed microstructures in CPaE membranes, which is characterized by the presence of water boundaries at its certain range of thickness. Water boundary in turn manifested the differential mechanical feature of the inner and outer layers, which is characterized by a differentiated stress distribution. While outer layer shows hydroplasticity and behaves as thoroughly stress relaxation, inner layer contributes the viscoelasticity characterized by intensified elasticity and serves as a baseline stress level.

Intriguingly, the water responsive dual-mechanic responsive behavior observed in CPaE membranes represents a noteworthy conceptual advancement, which suggests an unknown watery viscoelastic state which has not been reported in polymer plastic systems. We try our best to elucidate this new discovery through a unique mechanical perspective.

- Tensile test measured the strain response of CPaE membranes to applied stress while humidity-controlled stress relaxation test measured the stress response to constant strain, providing insights into stress and strain distribution and deformation behavior.
- Dynamic mechanical temperature analysis (DMTA) further provides useful insights into the variation of mechanical properties during the water adsorption and desorption in CPaE membranes with different thickness, which serves as strong evidence of the formation of spatially regulated microstructures.

In addition, we also get an inkling in the polarized optical microscopy (POM) measurements. Please see our response to comment 3.

3. The stress distribution in the cross-section of water-trained and air-dried films may be observed by polarized optical microscopy to understand the different shape recovery pathways in different cases.

Response

We appreciate reviewer#1 for the constructive suggestion. Since the polarized optical microscopy (POM) measurements could not be conducted underwater, the observation of stress distribution in membranes is performed by using air dried samples. We have added related discussions in the supplementary information.

In page 7, 16 of the Revised Supplementary Information

11. Analysis on Polarized Optical Microscopy (POM) images for air-dried CPaE membranes

The air-dried CPaE membranes were training in water for 3min and 3 hours before the observation on Polarized Optical Microscopy (POM) (fig. S9). The cross section of the two specimens look identical and transparent under natural light. Under polarized light, however,

the specimen (with 3 h water training) becomes colored because of its differentiated stress distribution in the inner and outer layers, while the specimen (with 3 min water training) completely merges into the black background.

fig. S9.

Optical Microscopy (OM) images shows the cross section of air dried CPaE membrane strips being made after 3 min (a) and 3 h (b) water training, respectively. Inset shows the enlarged Polarized Optical Microscopy (POM) images.

4. In Figure 3B, “Fracture energy” should be “Fracture work” or “Toughness”. “Fracture energy” generally describes the resistance to crack propagation and has a typical unit of “J m⁻²”.

Response

We appreciate reviewers#1 for pointing out this problem. To ensure clarity and precision, we have replaced the term “Fracture energy” to “Fracture work” in the main text.

5. In the stress relaxation tests, the authors wrote, “Starting at 95% RH, the initial stress of hydrated mid-CPaE significantly and quickly decreased from maximally 18.0 MPa to and

remained less than 0.1 MPa at 95% RH in the following cycles.” Such a substantial stress relaxation corresponds to the complete plasticization of the materials. However, even immersed in water for 72 h, the material is still very stiff and strong (Figure 3a). How to explain this difference?

Response

We appreciate reviewers#1 for the valuable comments. We apologize for the misleading descriptions on the discussion of stress relaxation tests and have made revisions to demonstrate this result clearly. The stress relaxation tests and tensile tests (in the case of 72h water immersion) used same hydrated specimens, all of which demonstrated stiff and strong mechanical properties. This can be verified by the results in cycle 1 of the stress relaxation tests. The outer layer of hydrated mid-CPaE should be near stress-free at 95% RH due to the binding of sufficient water molecules.

In page 10 of the Revised Manuscript

While the stress in hydrated mid-CPaE membranes at 95% RH decreased from maximally 18.0 MPa to 4.9 MPa, the elastic inner layer was major involved and retained a considerable portion of stresses. By lowering the RH to 30%, the stress recovered approximately to the original level, results in the majority of stress accumulation and turning the outer layer to weaker viscoelastic state. This indicated the presence of concealed spatially distributed microstructures characterized by an uneven distribution of strain while changing the RH¹³. Upon exposure to 95% RH again, water molecules abruptly penetrate into the membranes from the outer layer, and thus to relax the accumulated stresses to near 0 MPa rapidly. This further indicates that the adsorption and desorption of water molecules occurred in the same outer layer, which is a typical feature of reversible hydroplasticity. The statement of a distinctive spatial stress distribution in hydrated mid-CPaE membranes is strongly

evidenced by the fact that 1) The prominent difference in the stress relaxation behavior between the cycle 1 and subsequent cycles; 2) The Polarized Optical Microscopy (POM) results on the cross section of air dried CPaE membranes being made after 3 min and 3 h water training, respectively (fig. S9).

6. The author claimed that the water-responsive dual-mechanic functionalities is dominated by dynamical modulation of plasticity and viscoelasticity in the plastic with water training. However, the water-modulated plasticity and viscoelasticity of cellulose phenoxyacetate is not clearly demonstrated in the manuscript.

Response

We appreciate reviewers#1 for the valuable comments. We have added related discussions in the Revised Manuscript.

In page 11, 8 of the Revised Manuscript

Dynamic changes in the mechanical properties of CPaE membranes in watery and airy states, resulting from competing mechanical forces were demonstrated in Fig. 3E. During 3 h water training, the intensified elasticity characterizing the viscoelasticity of inner layer and the hydroplasticity of outer layer engaged in a fierce competition, with the former maintaining its elastic geometry and the latter restoring its original geometry. The inner layer serves as a baseline stress level while the outer layer showed thoroughly stress relaxation due to sufficient bounded water molecules (fig. S10). A delicate balance of plasticity and viscoelasticity manifested itself in an elastic shape memory response, presenting a watery viscoelastic shape memory. Upon air drying, new viscoelastic geometry was established in the outer layer due to the reversible hydroplasticity. Airy viscoelastic shape memory maintained when the outer layer's newly established viscoelasticity surpasses that of the inner layer. This shape memory

promptly restored to its original geometry upon water sorption. Notably, the viscoelasticity of the inner layer began to impede the development of hydroplasticity as water diffuses until it reaches its boundaries, thereby preserving the elastic geometry. To further elucidate the stress transfer and deformation change from such competitive mechanics, we visualized such nonlinear elastic behaviors on the whole timescale by using the modified viscoelastic-plastic model consisting of an elastic-plastic branch and a viscoelastic branch¹⁹ (fig. S11).

Fig. 3: **(E)**, Visual demonstration of watery and airy viscoelastic shape memory in CPaE membranes, distinguished by its spatially distributed mechanical feature.

7. Water content in the hydrated cellulose phenoxyacetate is an important parameter in determining the shape change behavior. What's the water content of the hydrated cellulose phenoxyacetate after immersing in water for different time? In Figure 2A, it is demonstrated that water molecules distribute inhomogeneously in the strip. Is there any way to detect the water distribution in the cellulose phenoxyacetate at various hydration time?

Response

We appreciate reviewers#1 for the valuable comments. We have conducted the experiment and added related discussions in the supporting information.

In page 7, 17 of the Revised Supplementary Information

12. Time-dependent water sorption dynamics in CPaE membranes

To understand the water sorption dynamics in CPaE membranes, time-dependent water sorption curve was recorded (fig. S10). The adsorbed water content increased from 0.084 ± 0.010 to 0.095 ± 0.014 as the immersion time increasing from 5 min to 3 h, and remained constant thereafter.

fig. S10.

Water Sorption Dynamics in CPaE membranes. Time-dependent water sorption curve in an ambient condition (16.7-22.6 °C, 46-65 % RH). Inset shows the static water contact angle on their surfaces. The scale bar is 1 mm.

Existing reports document cases where neutron radiography or Magnetic resonance imaging, combined with other techniques, are employed to visualize the water distribution in materials. Here are two typical examples for your reference.

Example 1

Neutron radiography and dielectric analysis under controlled RH (DEA-RH) were used to track moisture diffusion in untreated and nanocellulose-consolidated cotton canvases.

- Bridarolli, Alexandra, et al. *ACS Applied Polymer Materials* 3.2 (2021): 777-788.

Figure. a, T (%) vs time for cotton CNF-treated canvas together with four average neutron transmission images. b, (1–4) corresponding to different exposure times and RH values. Red to blue areas on the images indicate areas from high to low T (%), hence moisture content from low to high.

Example 2

Magnetic resonance imaging (MRI) was used to monitor water evolution and further quantify the correlation between programming, phase separation and recovery.

- Ni, Chujun, et al. *Nature* 622.7984 (2023): 748-753.

Figure. a, Evolution of MRI mapping for hydrogels with different programming times.

Unfortunately, these techniques suffer from considerable limitations in spatial resolution. Neutron scattering typically achieves resolutions of 80-110 micrometers in most reports, while MRI typically provides resolutions of about 1 millimeter. Moreover, a majority of cases are concentrated on water-enriched materials. Besides, measurements using these techniques cannot be achieved underwater. All of which collectively make it challenging to accurately detect the water distribution within CPaE membranes at various hydration time. We definitely acknowledge that the variable mechanical properties involved in the interaction between water and materials remains a complex and critical scientific issue. In this study, we have endeavored to utilize self-designed experiments and new analytical method to offer insightful perspectives for understanding our novel findings, rather than trying to be all-inclusive in pursuit of a definitive answer.

8. It is mentioned in Page 7, Line 132-136 that “Such water-responsive dual-mechanic functionalities could also be introduced into other polymer materials..., but not into polymers with too strong hydrophobic properties or without differentiated stress distribution (fig. S5 to 6).” More explanation needs to be provided.

Response

We appreciate reviewers#1 for the kind suggestion. We have provided necessary explanations in the Revised Manuscript.

In page 6 of the Revised Manuscript

This is due to the fact that too strong hydrophobicity can lead to two troubles: 1) Water is losing its plasticizing capacity necessary to change the geometry shape of CPaE membranes; 2) The water diffusion from the surface of CPaE membrane into interior is severely restricted, thereby impeding modulation of its inner layer's viscoelasticity. In addition, spatial differentiated stress distribution is imperative for water-responsive dual-mechanic functionalities of CPaE membranes, which is characterized by the selective formation of spatial chain relaxation. Through 3 h water training, a fraction of stress retained via the inner layer's viscoelasticity, where elastic memory locks the stress field. In the meantime, another fraction of stress rapidly relaxed to near 0 MPa via outer layer's plasticity. By contrast, homogeneous spatial stress distribution results in a single water-responsive mechanic functionality, which is based on the outer layer's viscoelasticity.

Reviewer #2

In the work, "Water training initiates spatially regulated mesostructures with competitive mechanics in hydroadaptive polymers" W. Chen et al reported the study of polymeric membranes with complex shape memories. The membranes can restore water-trained shape after sequential air training. A substantial part of the manuscript is devoted to the study of various elastic properties of these materials. However, some issues should be resolved before I can recommend this manuscript for publication.

Response

We very much appreciate Reviewer #2 for supporting the publication of our research work, and for raising important points need to be addressed. We have tried our best to made revisions in the main text accordingly. In addition, we have provided point-to-point response to the valuable concerns below.

1. The motivation for this work is unclear to me. The authors started with the problem of global plastic pollution, unfortunately, for me, it remains unclear how these materials could help in solving the problem.

Response

We appreciate reviewers#2 for providing the opportunity to reclarify our motivation for this work. In response to this comment, we have thoroughly revised the introduction to better highlight the novelty from a fundamental view.

In page 2 of the Revised Manuscript

Nature fascinates with its mechanical adaptive capabilities, as evident in the remarkable stimulus-responsive behavior of various materials¹⁻³. Among them, soft tissues such as cartilage and skeletal muscle stand out, as they not only offer mechanical support but also

actively reconfigure themselves, adapting to their environment and performing specialized functionalities⁴. When mimicking the mechanical adaptability found in nature, a majority of the artificial materials rely on soft matrices characterized by low moduli (of kilopascals (KPa) and megapascals (MPa)). Examples of such materials include hydrogels, ionogels, elastomers, and semi-crystalline resins⁵⁻⁷. Mechanically adaptive materials with high and tunable modulus in the gigapascal (GPa) range are rare. A prime example are high-performance nanocomposites inspired by natural load-bearing structures, which benefit from the integration of mechanical adaptivity⁸.

Here, we discovered previously unknown microstructures in polymers, actively regulated in space based on their distributed mechanical properties. This was achieved through a novel water training strategy applied to cellulose phenoxyacetate (CPaE). Such a counterintuitive water training strategy can adjust specific chronological parameters for modulating the viscoelasticity and plasticity. To achieve water-responsive dual-mechanic functionalities, membrane strips of CPaE polymers are trained in water for various time periods, followed by reprogramming to air shapes prior to their immersion in water. To understand the emergent microstructure-property relationships, we attempt to assess the manipulability of the competitive mechanics and to elucidate the general principle of differential mechano-responsiveness in terms of the water permeation and plasticization at the molecular scale, which regulates the stress changes and corresponding deformation at a structural scale.

2. I suggest expanding the discussions on the origin of the revealed memory-shape properties.

Response

We appreciate reviewers#2 for the suggestion. We have added the demonstration in the Revised Manuscript.

In page 11, 8 of the Revised Manuscript

Dynamic changes in the mechanical properties of CPaE membranes in watery and airy states, resulting from competing mechanical forces were demonstrated in Fig. 3E. During 3 h water training, the intensified elasticity characterizing the viscoelasticity of inner layer and the hydroplasticity of outer layer engaged in a fierce competition, with the former maintaining its elastic geometry and the latter restoring its original geometry. The inner layer serves as a baseline stress level while the outer layer showed thoroughly stress relaxation due to sufficient bounded water molecules (fig. S10). A delicate balance of plasticity and viscoelasticity manifested itself in an elastic shape memory response, presenting a watery viscoelastic shape memory. Upon air drying, new viscoelastic geometry was established in the outer layer due to the reversible hydroplasticity. Airy viscoelastic shape memory maintained when the outer layer's newly established viscoelasticity surpasses that of the inner layer. This shape memory promptly restored to its original geometry upon water sorption. Notably, the viscoelasticity of the inner layer began to impede the development of hydroplasticity as water diffuses until it reaches its boundaries, thereby preserving the elastic geometry. To further elucidate the stress transfer and deformation change from such competitive mechanics, we visualized such nonlinear elastic behaviors on the whole timescale by using the modified viscoelastic-plastic model consisting of an elastic-plastic branch and a viscoelastic branch¹⁹ (fig. S11).

Fig. 3: (E), Visual demonstration of watery and airy viscoelastic shape memory in CPaE membranes, distinguished by its spatially distributed mechanical feature.

3. There are several grammatical and stylistic errors throughout the main text. A thorough revision of language and grammar is necessary.

Response

We appreciate reviewers#2 for the important feedback. We have made a thoroughly revision of the main text to address any grammatical and stylistic errors, ensuring that the manuscript meets the highest standards of readability and coherence.

4. I haven't found the input files for the quantum chemistry calculations. Please provide them as supplementary materials.

Response

We appreciate reviewers#2 for the kind suggestion. We have included them in the source data.

5. In Fig. 3E the authors demonstrate the model diagram, however, I didn't calculations find the calculations in the text. How one can verify the applicability of the model?

Response

We concur with reviewers#2 regarding the concerns on the applicability of the mechanical models. In this study, the primary scientific issue centers on understanding the specific mechanical and microstructure evolutions occurring within the material during water training, leading to notable variations in its responsiveness to water over different training durations.

Through the construction of mechanical models, it is indeed helpful to better understand and quantify the contributions of the dynamic viscoelasticity and plasticity (*Duan J, et al. Physical Review Materials, 2023, 7(1): 013601*), as well as the interesting mechanical phenomenon, such as strain hardening. In this regard, we have done a lot of investigation with the models on amorphous polymer system.

For example,

- *Clarijs, et al. Journal of Polymer Science Part B: Polymer Physics 57.15 (2019): 1001-1013;*
- *Arrieta, et al. Mechanics of materials 68 (2014): 95-103;*
- *Meijer, et al. Macromolecular Chemistry and Physics 204.2 (2003): 274-288;*
- *Hem, Jérôme, et al. Macromolecules 55.20 (2022): 9168-9185;*
- *Chui, et al. Macromolecules 32.11 (1999): 3795-3808;*
- *Vogt, Bryan D. Journal of Polymer Science Part B: Polymer Physics 56.1 (2018): 9-30.*

It is worth to mention that the research work titled "Hydrated Solids" (*Harrellson, et al. Nature 619.7970 (2023): 500-505.*) has offers valuable insights by introducing a novel hydroelasticity model utilizing spores as experimental subjects.

Regrettably, the interactions between water and polymers remains a considerable intricate process up to now. Our efforts have not yielded a sufficient dataset to construct precise models capable of elucidating our novel findings. Consequently, our primary focus has not been on

modeling endeavors, but rather on adapting existing models and providing qualitative descriptions to facilitate the illustration of our conceptual understanding of the process. Considering its new perspective in elucidating the stress transfer and deformation change, we opted to retain it and simply relocate it to the supplemental Information.

In page 18 of the Revised Supplementary Information

fig. S11.

Schematic diagram for the time-dependent nonlinear elastic response undergoing continuous force loading using modified viscoelastic-plastic model.

REVIEWERS' COMMENTS

Reviewer #1 (Remarks to the Author):

I would like to support the publication of this manuscript

Reviewer #2 (Remarks to the Author):

I have no further comments.